# Effective TCP Flow Management Based on Hierarchical Feedback Learning in Complex Data Center Network

**DOI:** 10.3390/s22020611

**Published:** 2022-01-13

**Authors:** Kimihiro Mizutani

**Affiliations:** 1Department of Informatics, Kindai University, 3-4-1, Kowakae, Higashi-Osaka 577-8502, Osaka, Japan; mizutani@info.kindai.ac.jp; 2Cyber Informatics Research Institute, Kindai University, 3-4-1, Kowakae, Higashi-Osaka 577-8502, Osaka, Japan

**Keywords:** reinforcement learning, TCP incast problem, bandwidth optimization

## Abstract

Many studies focusing on improving Transmission Control Protocol (TCP) flow control realize a more effective use of bandwidth in data center networks. They are excellent ways to more effectively use the bandwidth between clients and back-end servers. However, these schemes cannot achieve the total optimization of bandwidth use for data center networks as they do not take into account the path design of TCP flows against a hierarchical complex structure of data center networks. To address this issue, this paper proposes a TCP flow management scheme specified a hierarchical complex data center network for effective bandwidth use. The proposed scheme dynamically controls the paths of TCP flows by reinforcement learning based on a hierarchical feedback model, which obtains an optimal TCP flow establishment policy even if both the network topology and link states are more complicated. In evaluation, the proposed scheme achieved more effective bandwidth use and reduced the probability of TCP incast up to 30% than the conventional TCP flow management schemes: Variant Load Balancing (VLB), Equal Cost Multi Path (ECMP), and Intelligent Forwarding Strategy Based on Reinforcement Learning (IFS-RL) in the complex data center network.

## 1. Introduction

The architecture of large-scale data center networks is optimized for providing many cloud services [1,2,3]. As the data center network handles the massive number of TCP flows for supporting high bandwidth use among network components (e.g., routers, switches, and servers), many TCP control schemes specific to data center architecture have also been proposed [4]. Figure 1 shows the structure of the conventional data center network (e.g., Fat-tree), and the process for establishing TCP flows among network components. In the Fat-tree architecture, the data center network consists of four layers of components: the Core, Aggregator, Edge, and Server layers. A certain layer’s component has multiple physical connections for the lower layer’s components, which are called child components, and a component receiving a TCP flow establishment relays it to a child component. In detail, each component holds TCP flows sent from the clients temporally, and relays them to bottom servers by iterative TCP flow establishment to each child components. To realize high bandwidth use in Fat-tree, redundant paths are ready to network components, and each component controls TCP flows with proper use of the paths [1]. Other data center architectures such as B-cube and DCell also fulfill the high bandwidth use with suitable TCP flow control [5,6], and their packet controls on a flow are implemented into programmable network components such as OpenFlow switches and intelligent routers [7,8,9]. Despite adopting the excellent data center architecture, the developers of the data center network have suffered from a TCP incast problem, which leads the degradation of TCP throughput [10]. In detail, a network component has a queue for preserving TCP segment data (contained into a TCP flow), and shares it for multiple TCP flows. If a lot of short flow (e.g, lightweight messages) arrives at the network component, the queue is overflowed and the arrival packets are dropped. This TCP incast problem probably occurs in the internet in the same way as a data center network. However, the communication quality on the Internet is not guaranteed (i.e., best effort communication), so the temporal occurrence of TCP incast is not critical for the users. On the other hand, a data center network sustains numerous cloud services with their requested service-level agreement (SLA), therefore, the communication delay caused by TCP incast probably violates the SLA and impairs the reliability of the data center network. To prevent the occurrence of TCP incast, a large number of TCP flows must be processed with several transport layer’s technologies on the data center architecture.

Avoiding TCP incast, the approaches of adopting Data Center TCP (DCTCP) focus on a rate adjustment, which means the packet sending interval adjustment among network components [11,12,13]. However, its effectiveness has a limitation for optimizing the entire TCP flow control in the data center network as its rate adjustment is executed on single TCP connection between ingress and egress components (i.e., peer to peer). Alternatively, using multipath TCP (MPTCP) scheme, a network component can forward packets through multiple TCP connections in parallel. It realizes load balancing of packet forwarding processes on the multiple TCP connections, and it contributes to the high bandwidth use and transferring a large amount of data among network components [14,15,16]. However, a policy for selecting multiple paths to establish TCP flow relies on ECMP (Equal Cost Multi Path) protocol, which allocates the packet forwarding processes to multiple TCP paths equally. In addition, similar to DCTCP, MPTCP can only optimize packet forwarding on the target paths.

To achieve effective bandwidth use in the entire data center network, this paper proposes TCP flows establishment scheme based on hierarchical feedback reinforcement learning. The feature and advantages of the proposed scheme are as follows.

For recognizing the entire network condition, the proposed scheme uses the hierarchical information which has been fed back from bottom to top components.Learning said information, the proposed scheme makes the empirical rule of efficient TCP flow establishments, and tries to establish TCP flows for optimizing TCP throughput in the entire data center network.Applying deep reinforcement learning approach into the learning algorithm, the proposed scheme can handle a massive number of TCP flows effectively.

The feedback information can be included in general TCP information exchanging so that the redundant communication (i.e., communicating fed back information) can be reduced. In addition, using deep reinforcement learning implementation bundled into famous calculation library (e.g., Chainer [17]) can also reduce the computational cost for applying intelligent TCP control. Therefore, the proposed scheme can realize the TCP throughput optimization with lower overhead. The rest of this paper is structured as follows. Section 2 introduces the related work concerning TCP flow management in data center networks. Section 3 explains hierarchical reward feedback learning and discusses simulation results of the proposed scheme in Section 4. Finally, Section 5 concludes this paper’s body and presents the future work.

## 2. Related Work

There are many approaches to avoid the incast and achieve effective bandwidth use in a data center network, which roughly consist of two groups described below.

The first approach tries to adjust a TCP flow rate control scheme such as DCTCP, which achieves high bandwidth and low latency by adjusting TCP rate by using Explicit Congestion Notification (ECN). Detecting an expansion of a packet queue length in its own packet forwarding function, a network component begins attaching ECN onto a TCP packet. After that, a network component receiving the packet-attached ECN negotiates with the sender to reduce the TCP rate for preventing the occurrence of TCP incast. The customized DCTCP for improving the bandwidth use in the entire data center network also proposed [12,13]. These schemes adopt a central information management server collects the network condition of the entire data center network, and determines TCP flow control policies in each network component. Alternatively, the decentralized approaches for tuning TCP flow rate are also proposed [18], which estimates the network condition by using the statistical message passing method, and deals with TCP incast as soon as possible. While TCP flow control based on ECN is proposed, FAST-TCP and TCP Vegas can realize effective TCP rate control not using ECN [19,20]. These schemes can control TCP flow rate by considering the queuing delay among network components, which enables them to estimate the states of TCP flows without ECN.

The second approach considers how to establish TCP connections. Valiant load balancing (VLB), which uses an oblivious flow control strategy, recursively forwards packets or TCP flows for the lower layer network components at random. Thus, VLB can handle TCP flow establishment independent of link capacities and TCP flow lengths. Alternatively, there are also multiple ways to implement randomized load balancing among multiple TCP flows. For example, MPTCP realizes multiple TCP flows establishments for sending a large number of packets in parallel, whose connections used in MPTCP are found by ECMP protocol [14,15]. The related works concerning MPTCP propose the congestion flow migration and switching multiple TCP flow control algorithms on multipath TCP flows [16,21]. The former changes the congested path of TCP to another one dynamically so that TCP incast occurrence can be dealt quickly. The latter switches multiple congestion control algorithms depending on network conditions. These schemes adjust TCP flow rate among multiple paths. Applying intelligent statistical approaches for MPTCP’s rate control has also been proposed, which improves the robustness for changing network condition dynamically [22,23]. The former approach adjusts TCP rates across multiple paths by using deep reinforcement learning. The latter approach can deal with drastic traffic changes that could not be solved by the former approach. There are also schemes that intelligently determine TCP path selection without using ECMP and MPTCP [9,24,25,26]. These schemes determine which path to establish TCP by learning algorithms. In the algorithms, a learning agent (e.g., network component) tunes the determination policy to minimize Round Trip Time (RTT) or throughput of a TCP flow.

The first approach focuses on end-to-end TCP rate control on single TCP connection so that it contributes only to the use of end-to-end bandwidth. Moreover, it has no mention of how to establish a TCP path among network components; thereby, the effectiveness is limited for improving bandwidth use in the entire data center network. Alternatively, the second approach considers how to establish paths of TCP flows and improves the bandwidth use in entire data center network compared to first approach. However, several schemes based on MPTCP depend on ECMP for establishing multiple TCP flows. Therefore, the path selections become inefficient in complicated data center networks in which link capacities are different. The path selection schemes without using ECMP and MPTCP can realize effective bandwidth use even if link capacities are different. However, these schemes focus on end-to-end bandwidth use and cannot realize the bandwidth optimization in the entire data center network.

Against these works, this paper’s proposed scheme makes the empirical rule of efficient TCP flow establishments in “entire” data center network. In detail, each network component measures RTT and TCP throughput, and determines the TCP flow path with reinforcement learning policy with the measured metrics. Additionally, the measured metrics are hierarchically fed back to share among lots of components, and are used in each component’s reinforcement learning. Note that the reinforcement learning algorithm uses both the measured and the fed back metrics. This cooperative manner can realize bandwidth optimization in the entire data center network.

## 3. Design of TCP Flow Management Based on Hierarchical Feedback Learning

This section describes the details of the proposed flow control scheme by using a hierarchical feedback model. The proposed scheme consists of three modules, learning module, resource-control, and feedback modules, which are implemented into a network component. Figure 2 shows the procedure of the proposed scheme’s flow management in a network component. Suppose network component A has the physical connections to B and C (i.e., child components); the behavior of the network component A implementing the proposed scheme can be explained as follows.

Step 1:When network component A receives a client’s TCP flow at time *t*, the resource-control module in the network component A selects child component (i.e., the lower connected component) to establish the TCP flow. This selection is determined by loading the connection states from the learning module. After this selection, network component A establishes a TCP flow to network component B.Step 2:Next, network component A receives the feedback information containing child network components’ network state (e.g., RTT and the number of held TCP connections) from network component B through a feedback module. If a previous TCP establishment causes TCP incast, the Round Trip Time (RTT) increases exponentially so that network component A learns its situation with the fed back information sent from network component B.Step 3:After a prescribed period of time elapses, if network component A detects the same situation observed at time *t* (in time t+a), the learning module predicts that TCP flow establishment for network component B causes TCP incast. Then, the resource-control module of network A does not establish TCP flow to the network component B; instead, the resource-control module establishes TCP flow to network component C.

To recognize the entire network condition, the network components implementing the proposed scheme feedback the network states from the bottom to the top components, hierarchically. Then, a higher layer component recognizes the lower layer’s network states and tries to improve the TCP flow establishment policy based on deep reinforcement learning.

The following subsections explain the behaviors of the three modules by using the mathematical data center model, and introduce the details of the implementation procedures.

### 3.1. Data Center Model

Suppose all network components in a data center network can control TCP flows from a client such as [27,28]. This subsection introduces a data center model that satisfies the following assumptions. Figure 3 shows an outline of the model.

A data center network consists of |N| network components, which figures hierarchical architecture, and the height of the hierarchy is *M*, noting that a network component in zero height indicates an end-host server.A network component i∈N has physical connections to network components below it, which are called child components of *i*, and the set of the child network components of *i* is expressed as $C_{i}$ noting that the network component $k\in C_{i}$ also includes other $C_{j}$ (i≠j), for example, network component 1 in Figure 3 has two upper links for two components located in higher layer (i.e., *i* and *j*).The number of established TCP flows from network component *i* to $k\in C_{i}$ is denoted as $F_{k}^{i}$.

The next subsection provides the basic concept of reinforcement learning used in the learning module, and indicates how to apply it to the module.

### 3.2. Reinforcement Learning

In reinforcement learning, a learning agent transits state $s_{t}$ from state $s_{t+1}$ by action $a_{t}$. Then, the learning agent obtains a reward $R_{s_{t},s_{t+1}}^{a_{t}}$ and adds the reward to the value $Q(s_{t},a_{t})$ in each state. Through the reward calculation, the learning agent aims to gain the maximum reward in each state. The learning agent finds the optimal action through the state transitions. When the learning agent transitions from $s_{t}$ to $s_{t+1}$ by action $a_{t}$, $Q(s_{t},a_{t})$, which is the value of the action $a_{t}$ in the state $s_{t}$, is updated as follows:$$\begin{array}{ccc} \;Q(s_{t},a_{t})\;\; & \leftarrow & \;\;\;Q\left(s_{t},a_{t}\right)+\alpha \left[R_{s_{t},s_{t+1}}^{a_{t}}+\gamma \max_{a}Q\left(s_{t+1},a\right)\;\;-\;\;Q\left(s_{t},a_{t}\right)\right], \end{array}$$
where α(0<α≤1) is the learning rate and γ(0<γ≤1) is the discount factor. The learning rate is a weight regarding to what extent the latest information will override the old information. The agent does not learn anything when α is 0; on the other hand, it considers only the newest information when α is 1. The discount factor γ is a weight for the effect of future rewards. The agent considers only the current reward when γ is 0; on the other hand, it tries to find a high reward in the long term when γ is 1. To apply the reinforcement learning to the proposed scheme, action, reward, and state in a network component should be defined.

### 3.3. Learning Module

The learning module learns the relationship between TCP throughout and the number of the established TCP flows based on a reinforcement learning mechanism. Here, let us define an action as a network component selection for establishing TCP flow, hence, network component *i* establishes TCP flow for child components at time *t* by action $a_{i,t}$. Then, the network component measures the TCP flows throughput in all held TCP and calculates a reward based on it noting that the concrete selection mechanism is explained in the subsection of the resource-control module.

Let $s_{t}^{i}$ denote a state stored in a learning module of the network component *i*, meaning each number of established TCP flows for the child components, and the fed back state from the child components, whose formula is shown in following.
(1)$$s_{i,t}=\left(F_{1}^{i}\;,F_{2}^{i}\;,\;\cdots ,\;F_{|C_{i}|}^{i},\;F_{1}^{*},F_{2}^{*},\;\cdots ,\;F_{|C_{i}|}^{*}\right),$$
where the state element of $F_{k}^{*}\left(k\leq C_{i}\right)$ is the number of all TCP flows for the *k*-th child component, noting that the number of TCP flows from the component *i* is excepted. For the component *i* in Figure 3, $F_{1}^{*}$ indicates the number of TCP flows between the network component *j* and 1. If the element is not measured or contained, the network component *i* cannot recognize the network condition (i.e., TCP throughput) for the *k*-th child component correctly and the state transition of the reinforcement learning is violated. For example, suppose that *k*-th child component connects two parent components. One of the parent components measures the number of TCP flows for the child component, and also do the throughput. Then, the throughput depends on the number of TCP flows held by the other parent component. Using reinforcement learning for maximizing TCP throughput in such a network, the recognition of the relationship between TCP throughput and the number of TCP flows is important so that the above state expression is a reasonable one for observing the correct state.

This state looks like a suitable expression for observing the network condition, but the records of the above states for reinforcement learning become significantly larger as the number of both TCP flows and physical connections increases. To address this problem, the proposed scheme adopts a deep reinforcement learning approach to approximate the records of the state’s information with a neural network model [29,30]. In concrete terms, $Q(s_{i,t},a_{i,t})$ is approximated by $Q(s_{i,t},a_{i,t},W_{i,t})$ where $W_{i,t}$ is the reinforced neural network model of network component *i* until time *t*. When a neural network model $W_{i,t}$ held in the network component *i* is reinforced with an obtained reward, the following update function is executed,
(2)$$W_{i,t+1}\leftarrow W_{i,t}-\alpha \frac{dL}{dW_{i,t}}{\left\{L\left(W_{i,t}\right)\right\}}^{2}$$
where $L_{t}^{i}\left(W_{i}\right)$ is TD-error (Temporal Difference Error) calculated in network component *i* at time *t*, whose value should be minimized by reinforcement learning, and it can be expressed by the following expression.
(3)$$L\left(W_{i,t}\right)=R_{s_{i,t},s_{i,t+1}}^{a_{i,t}}+\gamma \max_{a}Q\left(s_{i,t+1},A_{i},W_{i,t}\right)-Q\left(s_{i,t},a_{i,t},W_{i,t}\right),$$
where $a_{i,t}$ and $R_{s_{i,t},s_{i,t+1}}^{a_{i,t}}$ denotes the action $a_{i}$ and reward function of above state transition. Repeating Formulas (Equation 2) and (Equation 3), the output of $W_{i}$ (i.e., $Q(s_{i,t},A_{i})$) becomes closer to ground truth so that reinforcement learning can be realized without TCP states’ records. As the calculation or obtain processes of $R_{s_{i,t},s_{i,t+1}}^{a_{i,t}}$ and $F_{k}^{*}$ are executed in resource-control module and feedback module, respectively, the learning module tunes the learning model by gaining their information. The following subsections detail the processes in each module.

### 3.4. Resource-Control Module

A resource-control module of the network component *i* establishes a TCP flow for its child component. Then, the module reads $Q(s_{i,t},A_{i})$ from the learning module, and selects an action $a_{i,t}$ from a set of actions $A_{i}$ by considering at $Q(s_{i,t},A_{i})$ time *t*. In detail, the selection policy adopts the soft-max method based on Boltzmann distribution [31], and the probability of an action $a_{j}\left(j\leq \right|C_{i}\left|\right)$ is the following formula,
(4)$$prob\left(a_{j}\right)=\frac{e^{Q(s_{i,t},a_{j})}}{\sum_{j=1}^{|C_{i}|}e^{Q(s_{i,t},a_{j})}}.$$

From the above formulation, the higher the value $Q(s_{i,t},a_{j})$ is, the selection probability of the action $a^{j}$ is exponentially higher. At time *t*, the network component *i* executes an action $a_{i,t}$ or multiple actions by the soft-max method $\pi_{i,t}$, and tries to establish TCP flow for *j*’s child component.

Establishing the TCP flow, the resource-control module calculates a reward $r_{i}$, which is calculated by reward function $R_{s_{i,t},s_{i,t+1}}^{a_{i,t}}$ expressed by following,
(5)$$R_{s_{i,t},s_{i,t+1}}^{a_{i,t}}=\beta \frac{ST_{j}^{i}}{RT_{j}^{i}\sqrt{Loss}}+\left(1-\beta \right)r_{j},\;\;\;\;\;\left(a_{i,t}=a_{j}\right).$$

This reward function is based on a TCP throughput formula, which estimates the degree of TCP congestion with the discriminant model for TCP incast [19,20,32]. Suppose the network component *i* establishes TCP flow to *j*-th child component by action $a_{i,t}$ (i.e., $a_{j}$). $RT_{j}^{i}$ and $ST_{j}^{i}$ denote RTT and the shortest RTT between the network component *i* and *j* when a TCP flow is established between them. With these parameters, the resource-control module calculates the first term on the left side at first.

In addition, $r_{j}$ denotes the reward which is received from the *k*-th child component through the feedback module’s action. The reward is calculated in the child component with reward function $R_{s_{j,t},s_{j,t+1}}^{a_{j,t}}$. The reason why the child component’s reward is contained in the parent component’s reward is that the reinforcement learning control aims to consider the network performance (i.e., TCP throughput and RTT) of the lower layer on the data center network. The parameter β(0<β<1) is the weight for balancing throughput for child component and lower component’s throughput. As the value of β is lower, its reward function is weighted to lower network components throughput and aims to lower layers’ throughput optimization. Figure 4 shows the abstract of this reward feedback process that the network components [A, B, C, D] on the TCP flow paths calculate the feedback rewards, respectively.

This reward function (Equation 5) decreases as response time among lower layer network components (i.e., child component) increases, due to reasons such as TCP incast, so that it is calculated by considering the lower layer’s throughput. Both terms in reward function (Equation 5) are input to the learning module, and the learning module reinforces the neural network model through Formulations (Equation 2) and (Equation 3).

### 3.5. Feedback Module

This module enables a network component to feed its state and a calculated reward to the parent network component. In the above cases, the network component $j(j\in C_{i})$ provides the calculated reward $r_{j}$ and the total number of TCP flows $F_{j}^{*}$ excepting the number of TCP flows from component *i*. By continuously feeding back the states and rewards, the feedback process is executed hierarchically. Then, a network component can recognize TCP throughput below it, and control TCP establishments by considering hierarchical data center architecture. Note that the network component *i* records continuously feedback both child components TCP flows information $F_{1}^{*},F_{2}^{*},\;\cdot ,$ and $\;F_{|C_{i}|}^{*}$ and the hierarchical feedback rewards. Both the resource-control and the learning modules use the recorded data asynchronously for real-time TCP flow control.

### 3.6. Learning Algorithm of the Proposed Scheme

Algorithm 1 shows the above-mentioned learning control process. At first, $W_{i,t}$ must be initialized, and start to control TCP flow with a measured state. For building input data, the network component *i* records continuously feed back both child components TCP flows information $F_{1}^{*},F_{2}^{*},\;\cdot \cdot \cdot ,\;F_{|C_{i}|}^{*}$ and the hierarchical feedback rewards. The record is used in determining a state. Extracting the up-to-date child TCP flow information, the network component *i* builds a state $s_{i,t}$, and handles it as the input for the proposed scheme. With the input state, the next action $a_{i,t}$ is determined by the soft-max method $\pi_{i,t}$ in the resource-control module, noting that $a_{i,t}$ establishes TCP flow to component *j*. Waiting for receiving the hierarchical reward, the network component *i* combines both the received reward and the reward calculated by itself. After that, the network component *i*’s learning module updates the reinforced learning model $W_{i,t}$ with TD-error $L_{i,t}$, calculated using the reward, and completes these processes.
**Algorithm 1** The hierarchical feedback learning control of network component *i***Inputs:**      The measured state $s_{i,t}$,      learning rate α,      balanced factor β, and discount factor λ.**Initialize:**      deep reinforcement learning models $W_{i,t}$# Obtain next action and execute it.$a_{i,t}\leftarrow \pi_{i,t}$# Suppose $a_{i,t}$ is establishing the TCP flow for component *j*.# Calculate reward function with Measured RTT.$R_{s_{i,t},s_{i,t+1}}^{a_{i,t}}$ = $\beta \frac{ST_{j}^{i}}{RT_{j}^{i}\sqrt{Loss}}$# Re-calculate reward function with the feedback reward.$R_{s_{i,t},s_{i,t+1}}^{a_{i,t}}$ += $\left(1-\beta \right)r_{j}$# Calculate TD-error$L\left(W_{i,t}\right)=R_{s_{i,t},s_{i,t+1}}^{a_{i,t}}+\gamma \max_{a}Q\left(s_{i,t+1},A_{i},W_{i,t}\right)-Q\left(s_{i,t},a_{i,t},W_{i,t}\right)$# Update $W_{i,t}$ to $W_{i,t+1}$$W_{i,t+1}\leftarrow W_{i,t}-\alpha \frac{dL}{dW_{i,t}}{\left\{L\left(W_{i,t}\right)\right\}}^{2}$t←t+1

### 3.7. Summary of the Proposed Scheme’s Behavior

Figure 5 shows the concrete behavior of the proposed scheme’s TCP flow establishment. At time *t*, a network component receives a TCP establishment from parent component (i.e., the connected higher layer’s component). Then, the resource-control module (RM) requests $Q(s_{i,t},A_{i})$ to the learning module (LM). The LM returns it by using $s_{i,t}$ consisted of the feedback parameters $F_{1}^{*},F_{2}^{*},\;\cdot \cdot \cdot ,\;F_{|C_{i}|}^{*}$ and the measured parameters $F_{1}^{i}\;,F_{2}^{i}\;,\;\cdots ,\;F_{|C_{i}|}^{i}$ noting that the feedback module (FM) obtains the input parameter and reward information from the child components at certain intervals. With $Q(s_{i,t},A_{i})$, the RM determines the next target for establishing TCP flow. Establishing TCP flow for the target, the RM measures the established TCP flow’s RTT, and inputs it for the LM. Finally, the LM combines the reward input to the RM and a feedback reward, and reinforces its own learning model with them. Repeating this process, the proposed scheme can realize an intelligent TCP flow establishment for improving bandwidth use in the entire data center network.

## 4. Simulation Results

This section explains how much the proposed scheme could reduce the occurrence of TCP incast with computer simulation. This simulation program is running on Ubuntu Linux server (CPU: Intel(R) Core(TM) i7-9700K 3.60 GHz, Memory: 32 GB, GPU: NVIDIA RTX 2080 super), and the proposed scheme’s learning algorithm is built by Chainer and ChainerRL of Python 3.6. ref. [17,33]. With these settings, 100 million TCP flows arrive at the data center and measure the probability of incast occurrence, RTT, and flow completion time.

Assuming a high-traffic data center, the simulator constructs two hierarchical data center topologies: Fat-tree and Tree, and provides storage service on it [1,2,5]. Figure 6 shows their topology that the tree topology is a non-complex architecture; on the other hand, the Fat-tree topology is a complex architecture, which establishes multiple paths for a destinated component. The hierarchical level is three and the numbers of network components are 10, 50, and 250 in each hierarchy, which derives from the general data center network model built on three levels components: the core, aggregate, and server layers’ components. Each layer’s network component, except for the core layer, has five connections to the lower layer’s components. The arrival rate of TCP flows from an outer network to a core router depends on the Poisson distribution, and the length of TCP flow (i.e., download time) depends on the exponential distribution, whose TCP flows traverse the data center network’s top to bottom [34,35,36]. The maximum numbers of TCP flows held in a network component are different from the located layer. A core/aggregate/server layer’s component can handle up to 100,000, 10,000, and 1000 TCP flows, respectively, noting that those capacities vary by about ±10% randomly. Each link is set RTT = [0, 0.001]; however, the RTT increases exponentially due to TCP incast. With these settings, the proposed scheme, VLB, ECMP, and IFS-RL are compared to their performance. The VLB randomly forwards TCP flows to lower components [37,38]. ECMP, adopting many MPTCP schemes, splits a flow into *n* subflows equally and forwards the split flows to lower components through different routing paths, noting the subflows path, which is calculated by OSPF [14,15,16]. IFS-RL is an intelligent forwarding strategy based on reinforcement learning, similar to the proposed scheme, which selects the flow establishment target (i.e., lower network component) to optimize the bandwidth use between network components [9]. The difference between IFS-RL and the proposed scheme is whether the learning algorithm, based on hierarchical information which has been fed back (i.e., state and reward), is applied or not. The proposed scheme and IFS-R accept five-layered Deep Q Network (DQN) and Double Deep Q Network (DDQN) to approximate $Q(s_{i,t},a_{i,t})$, noting that each layer’s input size corresponds to the first layer’s one. DQN and DDQN are reinforcement learning algorithms using Neural Networks, which are applied to implement the functions (formulas) of (Equation 2) and (Equation 3). Note that IFS-RL uses the same DQN and DDQN as the proposed scheme, however, does not consider the factors fed back. To deploy it to this simulation, both neural networks are tuned to 1000 epochs, and the learning parameters [α,β,γ] are set to [0.1,0.5,0.9].

### 4.1. Probability of TCP Incast Occurrence for Changing TCP Flow’s Pattern

Figure 7 shows the probability of TCP incast occurrence for changing the arrival rate of TCP flows among all network components on Tree and Fat-tree architecture. The changed target parameter λ determines the arrival rate of TCP flows by Poisson distribution. If λ becomes larger, the arrival of TCP flows is higher and causes TCP incast frequently. In all schemes on both architectures, TCP incast frequently occurs as the arrival rate of TCP flows is higher. However, both ECMP and VLB conduct the worse results as the performances are guaranteed on the stable hierarchical topology. Alternatively, the proposed schemes could reduce the probability of TCP incast occurrence of up to 20% compared with ECMP and VLB. Comparing to IFS-RL, the performance improvement rate decreases compared to ECMP and VLB. The proposed scheme and IFS-RL can learn the network condition, depending on the link capacity and arrival rate of TCP flows, and prevent the entire data center network from TCP incast.

However, it was found that the proposed scheme outperformed IFS-RL due to the realization of the accurate recognition of entire network condition. This trend is not different in both data center architecture, however the performance difference is more pronounced for the complex architecture compared to the non-complex architecture. Comparing DQN and DDQN performances, the proposed scheme based on DDQN is superior based on DQN. Generally, DQN is overoptimistic for finding the relationship and takes more statistical error in approximating $Q(s_{i,t},a_{i,t})$ than DDQN.

Figure 8 shows the probability of TCP incast occurrence for changing the length of TCP flows in both architectures. In this evaluation, the arrival rate fixed to 0.5 and the parameter λ of the exponential distribution of TCP flow length was changed. When the parameter becomes lower, the distribution of TCP flow length features a long tail, and the longer TCP flows arrive at the data center frequently (hence, TCP incast occurs frequently). This result shows that the proposed schemes saw lower TCP incast occurrences than the conventional schemes: ECMP, VLB, and IFS-RL whose result is similar to the previous result. The proposed scheme controls TCP flow establishment by considering TCP flow length, so that the probability of TCP incast occurrence is bound to about 20%, even if the flow length is longer. Therefore, as with the previous experimental results, there was a difference in performance for each scheme.

### 4.2. The Comparison of RTT, Throughput, and Flow Completion Time in Each Scheme

The next evaluations adopt the worst scenarios for each scheme, which were executed on the highest arrival rate of TCP flows. In this environment, RTT, throughput reduction, and flow completion time are measured by changing the flow length distribution value λ 0.1 (short flow arrivals) to 0.9 (long flow arrivals). Figure 9 shows the average of RTT in different data center architectures. In the case of short flow arrivals (label: Short), all schemes could take the average RTT up to 5 ms as TCP incast does not occur frequently, noting that an error bar indicates the 95th percentile. On the other hand, in the case of long flow arrivals (label: Long), TCP incast occurs more frequently than the case of short flow arrivals, as more packets occupy the entire data center network. Therefore, the RTT becomes longer and it took up to 8 ms. Focusing on the different architectures’ results, RTT of all schemes is the shorter in non-complex architecture than in complex architecture. In detail, the difference in average RTT in both architectures was about 2–3 ms. Additionally, it was found that the proposed scheme contributed most to the reduction in average RTT and its deviation.

Figure 10 shows maximum throughput degradation rate in different data center architecture. The meaning of both labels is the same as in the evaluation of RTT. In both architectures, the proposed scheme contributed to reduce the throughput degradation rate. In particular, the proposed scheme in the long case on complex architecture could realize the lowest degradation compared to the other schemes, at about 20% degradation.

Figure 11 shows the average flow completion time in different data center architecture. In both of the architectures, there was not much difference in completion time for the short flow in all schemes. However, the proposed scheme realized the shortest completion time in the other schemes, and it is about 10 ms in both architectures with the long case. IFS-RL also took a shorter completion time (about 15 ms), but not shorter than the proposed scheme. Focusing on 95th percentile of the completion time, the proposed scheme also realized shortest result.

### 4.3. Probability of TCP Incast Occurrence for Changing Learning Parameters

Figure 12 shows the effects of the learning parameters of the learning module. It measured the probability of TCP incast occurrence by changing learning rate α and discount factor γ in the complex data center architecture. The result indicated that the probability of TCP incast occurrence depends on both parameters. In the case of a lower learning rate, the probability becomes lower. In general, the lower learning rate enables the learning agent to obtain deep insight into the given environment; thereby, it can be concluded that the better learning control was realized compared with the higher learning rate case. On the other hand, as the discount factor γ became lower, the probability of TCP incast occurrence took a higher value. The discount factor γ determines the effect of future rewards. The agent considers only the current reward when γ is 0 and tries to find a high reward in the long term when γ is 1. Therefore, when the value of the discount factor becomes higher, stable flow control can be realized over a long period and the probability of TCP incast occurrence decreases.

Figure 13 shows the probability of TCP incast occurrence by changing the balanced factor β of reinforcement learning’s reward in the complex data center architecture. Adapting the largely biased value (e.g., 0.1 and 0.9), the proposed scheme took the higher probability of TCP incast than the case of adapting unbiased value (e.g, 0.5). The balanced factor β of reinforcement learning’s reward weights the TCP throughput feedback from the lower layer components. As the parameter β becomes higher, a network component largely reflects the TCP throughput measured by itself in its own reward. This means that the network component tries to maximize its own TCP throughput and does not aim to optimize the entire TCP throughput. On the other hand, as the parameter β becomes lower, a network component largely reflects the fed back TCP throughput (i.e., reward) in the own reward so that the network component does not maximize the throughput of the held TCP flows. From the result, setting the unbiased value to β is important to maximize the entire TCP throughput.

### 4.4. Overhead for Reinforcement Learning

The proposed scheme adopts the deep reinforcement learning approach to realize intelligent TCP flow control, thereby the computation delay occurs for each control. In detail, the delay depends on the performance of extracting $Q_{(}s_{i,t},A_{i})$ at each control. Using chainerrl library [33], both DQN and DDQN in the proposed scheme took about 3 ms and 4 ms to the extraction, respectively. This delay is not critical for improving the throughput and flow completion time. This is as the delay depends on the network component’s specifications (e.g., CPU and memory size), therefore it will be improved as the specification improved.

Besides the delay, the storage cost of the reinforcement learning model is also an overhead. Using chainerrl library, both DQN and DDQN models took tens of kilobytes at most. However, various reinforcement learning models must be preserved for dealing with the case of dramatically changing the arrival rate of TCP flows. For example, the reinforcement learning models optimized for a certain arrival rate (λ=0.1) cannot always archive the optimal TCP flow control in other case (λ=0.9). Additionally, a model selection algorithm must be implemented for the effective use of the reinforcement learning models. For dealing with the cases, the robustness of the reinforcement learning control must be improved.

## 5. Conclusions

This paper proposed a TCP flow control scheme for efficient bandwidth use in data center networks, which can be realized even if the data center architecture is complicated. In the simulation results, the proposed scheme showed that the scheme is more effective than other TCP flow management schemes (i.e., VLB, ECMP, and IFS-RL) in reducing the number of TCP incast occurrences, forwarding delay, and the flow completion time. This paper also evaluated the effects of learning parameters on its performance, and the results showed that its effectiveness can be enhanced by adjusting the learning parameters in a high-traffic data center network. For realizing more effective bandwidth use, the robustness improvement of the proposed scheme and its verification would be left as a future problem.

## Figures and Tables

**Figure 1 sensors-22-00611-f001:**
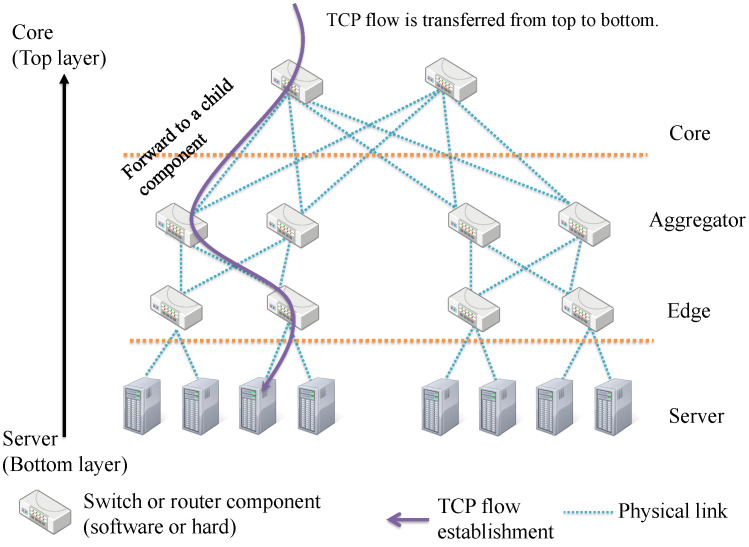
A structure of conventional data center network (e.g., Fat-tree).

**Figure 2 sensors-22-00611-f002:**
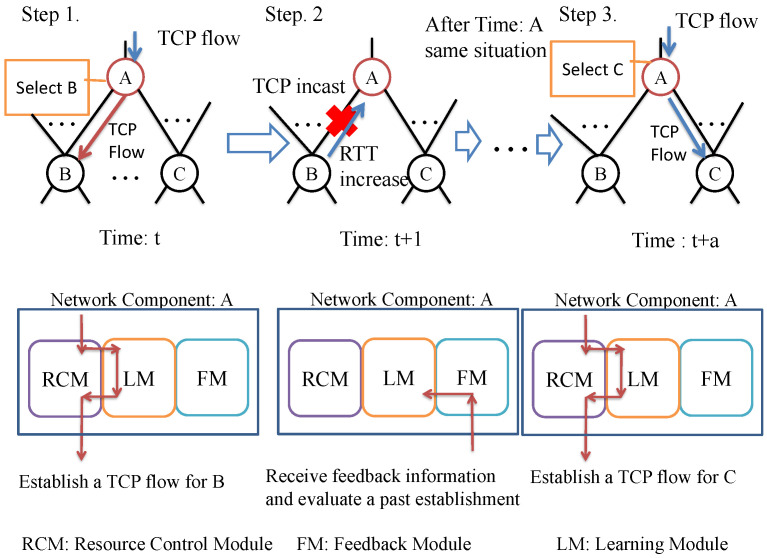
The architecture of the proposed scheme.

**Figure 3 sensors-22-00611-f003:**
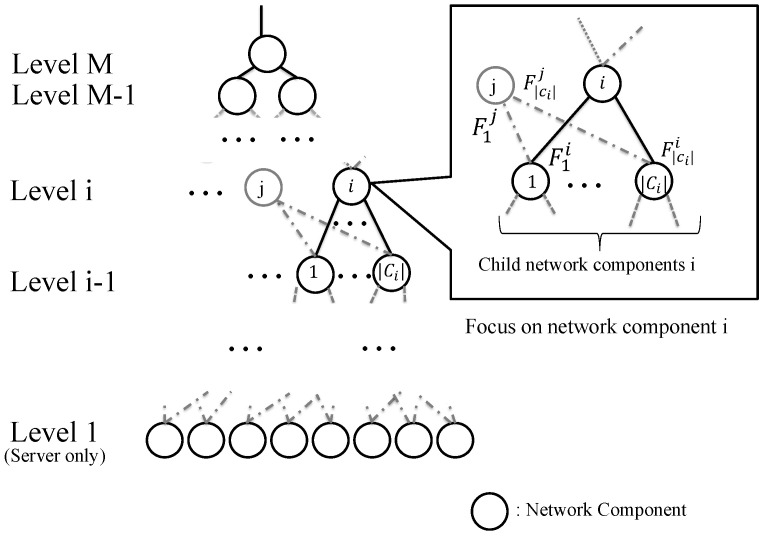
The outline of the data center model adopted in the proposed scheme.

**Figure 4 sensors-22-00611-f004:**
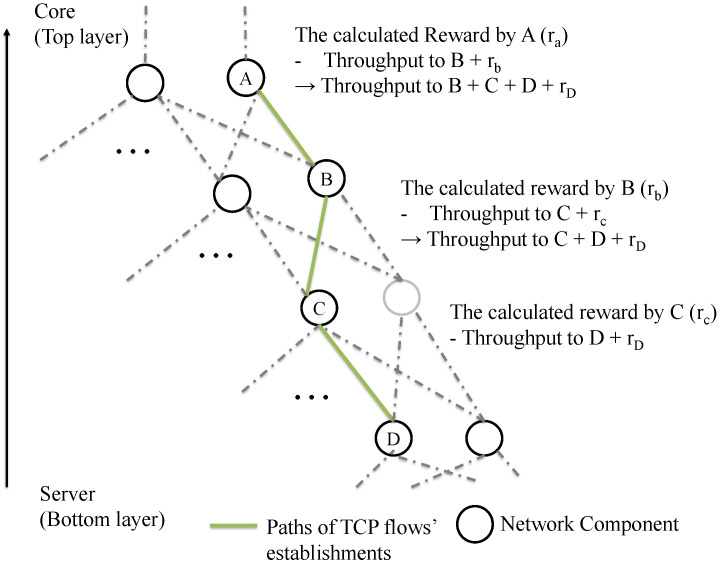
The abstraction of the reward feedback process (Hierarchical reward feedback).

**Figure 5 sensors-22-00611-f005:**
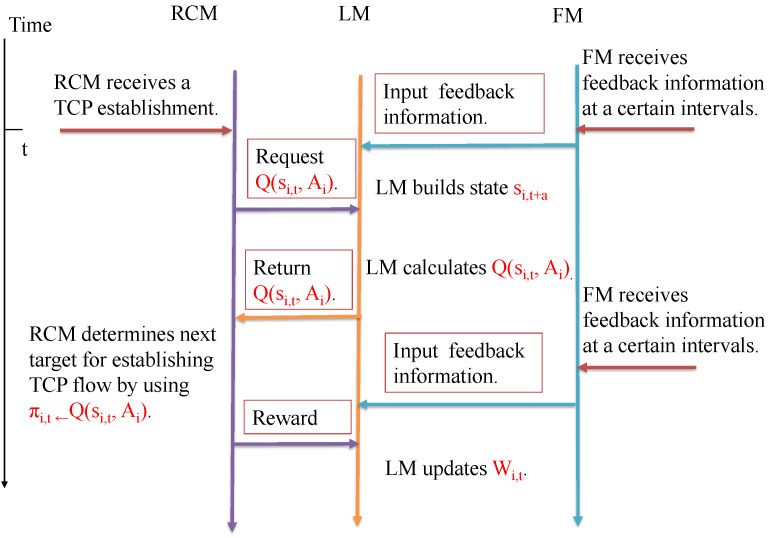
An sequence diagram of the proposed scheme’s behavior.

**Figure 6 sensors-22-00611-f006:**
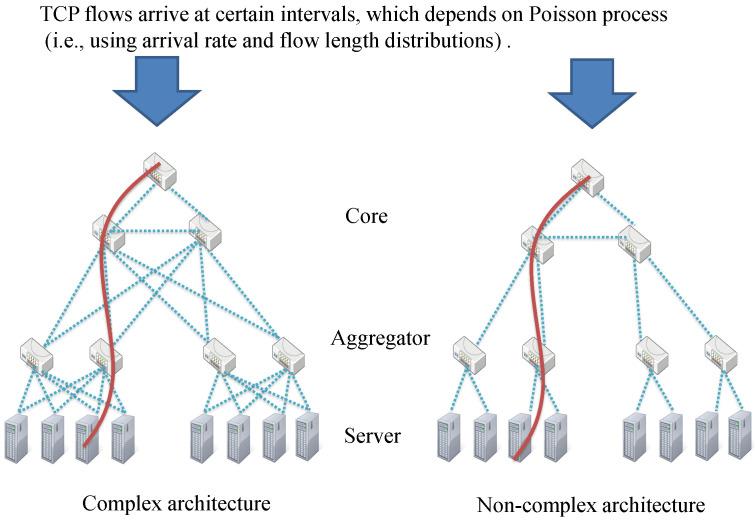
An illustration of the evaluation environment. In this evaluation scenario, Clients’ TCP flows arrive at both data center, whose traffic volume depends on the arrival rate and flow length distributions.

**Figure 7 sensors-22-00611-f007:**
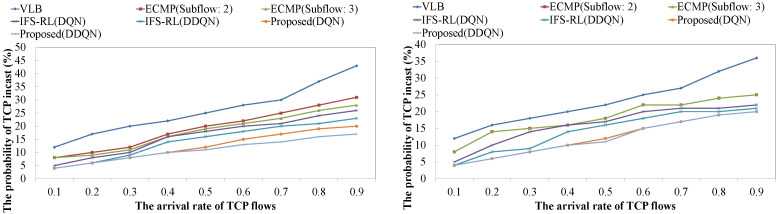
The probability of TCP incast occurrence against the arrival rate of TCP flows ((**Left**): the case for complex architecture, (**Right**): the case for non-complex architecture).

**Figure 8 sensors-22-00611-f008:**
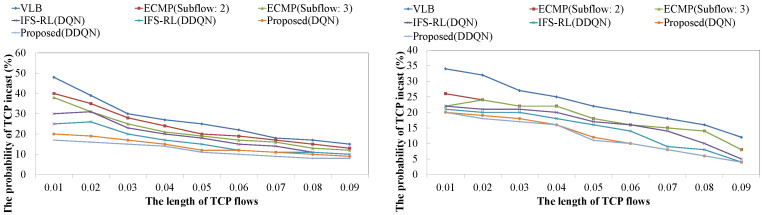
The probability of TCP incast occurrence against the length of TCP flows ((**Left**): the case for complex architecture, (**Right**): the case for non-complex architecture).

**Figure 9 sensors-22-00611-f009:**
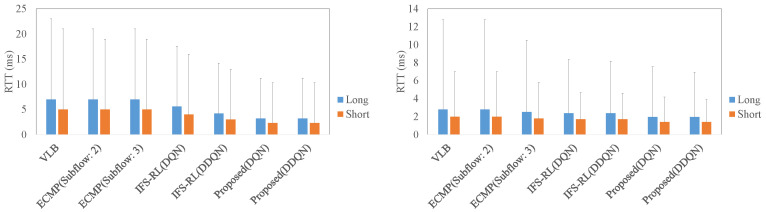
RTT against the length of TCP flows ((**Left**): the case for complex architecture, (**Right**): the case for non-complex architecture).

**Figure 10 sensors-22-00611-f010:**
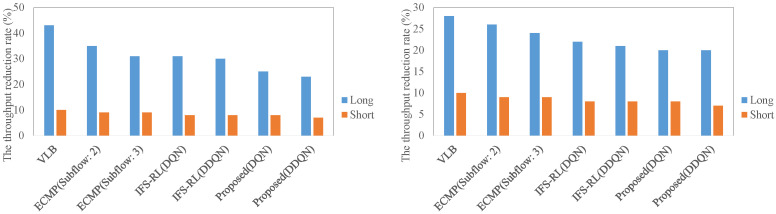
The maximum throughput degradation rate in different data center architecture ((**Left**): the case for complex architecture, (**Right**): the case for non-complex architecture).

**Figure 11 sensors-22-00611-f011:**
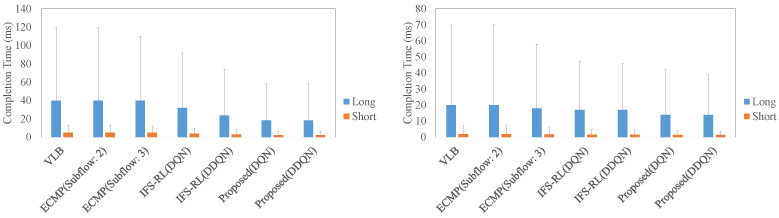
The average flow completion time in different data center architecture ((**Left**): the case for complex architecture, (**Right**): the case for non-complex architecture).

**Figure 12 sensors-22-00611-f012:**
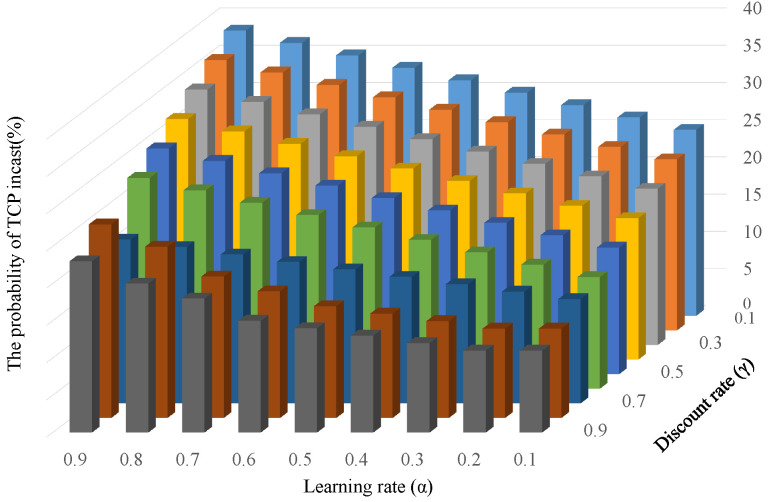
Probability of TCP incast occurrence against learning parameters: α,γ.

**Figure 13 sensors-22-00611-f013:**
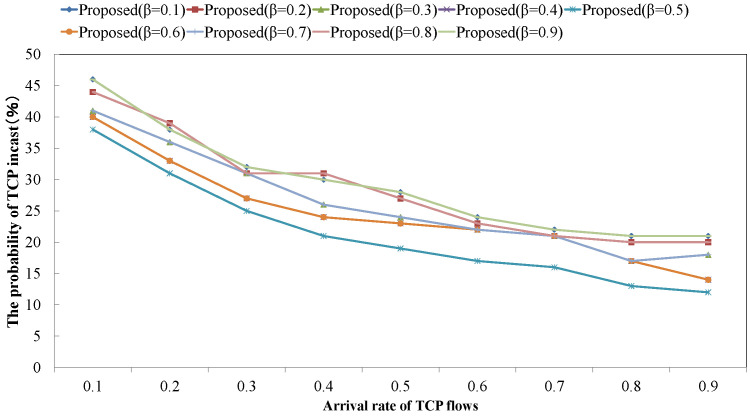
The probability of TCP incast occurrence against learning parameter: β.

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
