# Peer review of "Effective TCP Flow Management Based on Hierarchical Feedback Learning in Complex Data Center Network"

_sensors, 2022, doi:10.3390/s22020611_

Round 1

Reviewer 1 Report

The author proposed a TCP flow management scheme for data center networks. The idea has novelty; however, the writing is poor and hardly understandable in many places. Also, the authors have not presented enough results. Following are my comments:

  1. In the introduction section, the contributions are not clear enough. The contributions should be pointed out in bullet points.
  2. The writing is poor, the written English is very hard to understand, and contains numerous grammatical errors. The writing must be improved.
  3. A proper system flowchart should be presented.
  4. The experimental results are not sufficient. The compared parameter is uncommon. Results for throughput, delay, and flow completion time should be included.
  5. DQN and DDQN should be well defined at their first occurrence in line 311.
  6. An illustration of the proposed scenario should be included.
  7. Simulation should be done considering different scenarios and compared with recent advanced protocols, e.g., “Flow-aware adaptive pacing to mitigate TCP incast in data center networks” by Shaojun Zou at el.

Reviewer 2 Report

The authors have to address all of the below concerns carefully.

  • Abstract: The "I" (page1-line6) should not be used in scientific research (please check the entire paper). Some numeric findings for the proposed scheme should be added to the end of the abstract.
  • Keywords: We suggest that the authors should replace keywords such as “TCP flow management” and “Cloud network” because these keywords are already found in the review article title. It is better that they replace them with other keywords to increase the reach of the manuscript.
  • Introduction Section: What is incast problem? The author should explain it in more detail. Also, the background of the topic should be more extensive to explain the topic more comprehensively. Also, what are children components (page2-line42)? The introduction should be comprehensive and clear.
  • There are many terms that are used without being defined at first appearance such as TCP, VLB, ECMP, MPTCP, VL2, DCTCP, RTT … etc.
  • Related Work Section: The flaws and problems of the existing research should be written in more detail to highlight the gap.
  • Simulation results Section: Sentences in lines 314-317 should be written at the beginning of this section. What are the limitations of the proposed approach?
  • Conclusion Section: What are the future directions for this study?
  • Figures: All figures are drawn in high resolution. However, some figures shown before being summoned in-text such as Figures 2, 3, 5, 6 and 8. Figure 5 is not used in-text.
  • References list: The number of references is sufficient, but most of the references are old and require updating. Also, some references are not very relevant to the research topic. In addition, References should follow the MDPI-Sensors style. For instance, some search names in the reference list begin an uppercase letter for each word (such as [3], [9], [11] ... etc.) and others use only an uppercase letter in the first word (such as [1], [2], [4] … etc.), author should standardize style. Some references do not contain enough information as references [4], [22] …etc. The list of references requires extensive scrutiny by the authors.
  • English Writing: This paper requires moderate proofreading. There are some of grammatical, spelling and typos problems. Some sentences are very long and should be broken down into shorter sentences to be understandable. The authors have to thoroughly scrutinize the paper. Without professional, accurate and clear English, readers cannot understand the paper.

Reviewer 3 Report

In this paper, the author proposed a TCP flow management scheme specified a hierarchical complex data center network for effective bandwidth use. The proposed scheme dynamically controls the paths of TCP flows by reinforcement learning based on a hierarchical feedback model, which aims to obtain an optimal TCP flow establishment policy even if the network topology and the link states are more complicated.

In my opinion, the quality of the article is high and the discussed solution has great application. The paper is well written and the proposed scheme is clearly described. However, the following concerns need clarifications.

  1. The computation efficiency of the proposed method should be addressed.
  2. Explain more advantage of the suggested method.
  3. The quality of the figures is very low. Must be improved in the revised version.
  4. There are some minor grammatical errors in the text. The language of the paper needs a review.
  5. There are a large number of undefined abbreviations and are required to be there in full text like TCP, VLB and ECMP and many more.

Round 2

Reviewer 1 Report

The author has substantially improved the manuscript. It can be accepted in the current form.

Author Response

I wish to express our appreciation to the Reviewer for his or her insightful comments, which have helped us significantly improve the paper. I checked the paper body by using some English check tools and modified them.

Reviewer 2 Report

The author responded to most of our comments, however, there are still some minor comments that require a response and an address accurately.

    • Introduction Section: This comment still requires more details. What is incast problem? The author should explain it in more detail. Also, the background of the topic should be more extensive to explain the topic more comprehensively. Also, what are children components (page2-line42)? The introduction should be comprehensive and clear.
    • Simulation results Section: This comment still requires a response. What are the limitations of the proposed approach?
    • References list: Some references do not contain enough information as references [11], [36] …etc.
    • English Writing: This paper requires minor proofreading. There are some of grammatical, spelling and typos problems.
